# OpenReview forum: "Tree of Attacks: Jailbreaking Black-Box LLMs Automatically"
_NeurIPS.cc/2024/Conference — NeurIPS 2024 poster_

### Official Review · Reviewer_mTdE · 2024-07-07

**Soundness:** 4
**Presentation:** 4
**Contribution:** 3
**Rating:** 7
**Confidence:** 4

**Summary:**

Note: I have seen this paper before, and I have heard of the method. However, I have not previously read it in full, and I do not remember anything about the authors. So this should still be a fully blind review.

This paper introduces TAP -- a black-box method to develop attacks against LLMs. The attack process constructs a tree with a branching step based on refining a prompt using an LLM and a pruning step based on pruning attacks deemed off topic or assigned a low jailbreaking success score by another LLM. This method is thus able to search for attacks in a way that's automated and interpretable.

**Strengths:**

S1: I think that the overall contribution is valuable to the field. TAP is kind of simple and arguably a glorified data-augmentation technique, but I think that TAP clearly belongs in the red-teaming toolbox.

S2: One might criticize this paper for being an incremental improvement on PAIR, but I think that would be a pretty lazy criticism, and I don't buy that that would make this not valuable. I think that tree method is a good insight, and it doesn't have to be complicated to be useful.

**Weaknesses:**

W1: I think that the biggest potential problem with this work is the jailbreak evaluation method. Recent work like [https://arxiv.org/abs/2402.10260](https://arxiv.org/abs/2402.10260) has shown that LLM autograders for jailbreak success tend to be pretty problematic. Meanwhile, I'm very skeptical of a supposed 88% success rate against GPT-40 in Table 8. Overall, I don't think that what is being evaluated is probably an ideal proxy for "safety".

W2: The biggest limitation with TAP to me seems that its ability to help you find very novel jailbreaks seems very limited, and the initialization/prompting is doing a lot of work. I'd be appreciative of discussing this limitation or some more quantitative/qualitative analysis of it.

**Questions:**

See weaknesses.

**Limitations:**

See weaknesses.

---

> ### Author Rebuttal · Authors · 2024-08-07
>
> Thank you for your valuable feedback. We are happy that you appreciate our paper’s contribution and we take your concerns seriously. We respond to them below and will include our responses in the form of expanded discussions in the final version.
>
> **"Recent work…has shown…LLM autograders for jailbreak success…to be…problematic."** We agree that LLM autograders for jailbreak success can be inaccurate and this is an important problem for the field of jailbreaking. However, at the same time, we believe this is orthogonal to the contribution of this paper as the proposed method (TAP) is compatible with any existing or future automated methods for evaluating jailbreak success.
>
> **"I don't think that what is being evaluated [in Table 8] is probably an ideal proxy for "safety"."** Table 8 reports the success rate according to manual grading done by the authors. When grading, we anonymized the name of the method used to generate the jailbreak to avoid any inadvertent skew favoring our method. After this anonymization, we followed the guidelines in Wei et al. (NeurIPS, 2023) [52] to evaluate a successful jailbreak. We acknowledge that there can be many notions of safety. We would be happy to repeat this evaluation with other specific notions of safety and include a discussion on the current evaluation criteria and its limitations.
>
> **"ability to…find…novel jailbreaks seems…limited…I'd be appreciative of discussing this limitation…"** Thanks, we will include a discussion on this in a limitations section in the final version.

---

> > ### Comment · Reviewer_mTdE · 2024-08-09
> > **Thanks, I think the paper should be accepted**
> >
> > Thanks, not too much more to say. I basically think that we are on the same page. I'll hold at a 7. But I would encourage the authors to mention explicitly the problems with autograders. I would also add that mentioning any differences in eval methodology that might make the numbers in your tables and the numbers in other papers' tables apples-to-oranges.

---

### Official Review · Reviewer_2Heq · 2024-07-10

**Soundness:** 3
**Presentation:** 3
**Contribution:** 3
**Rating:** 5
**Confidence:** 4

**Summary:**

This paper presents an automated method, Tree of Attacks with Pruning, for generating jailbreak prompts to exploit vulnerabilities in LLMs. TAP uses a tree-of-thought reasoning approach to iteratively refine and prune candidate prompts, significantly reducing the number of queries needed to successfully jailbreak LLMs like GPT-4.

**Strengths:**

1. TAP's method of using tree-of-thought reasoning combined with pruning optimizes the search for effective jailbreak prompts, requiring fewer queries.

2. TAP demonstrates effectiveness across multiple LLMs and attack scenarios, including models with advanced protections like LlamaGuard.
The evaluations show TAP successfully jailbreaks most state-of-the-art LLMs for over 70%.

**Weaknesses:**

1. The format of this paper needs to be refined, such as Table 1,2,3.

2. The success of TAP heavily depends on the choice of evaluator LLM, with significantly reduced performance when using less powerful evaluators.

3. Much lower attack success rate on well aligned open-sourced model, Llama-2-7b-chat. Llama3 and gemma2 may be needed for evaluating the effectiveness of this method.

**Questions:**

See weakness part.

**Limitations:**

The running time of this method is long.

---

> ### Author Rebuttal · Authors · 2024-08-07
>
> Thank you for your feedback. We take it seriously and respond to your concerns below. We hope that based on our responses you will strengthen your support for the paper.
>
> **“format of this paper needs to be refined.”** Thanks, we will refine the format.
>
> **“The success of TAP heavily depends on the choice of evaluator LLM…”**  Yes, the success of our method (TAP) does depend on the choice of the evaluator. However, this is not specific to our work: both Yu et al. [59] and Chao et al. [12] use LLMs to evaluate whether jailbreaking attempts were successful. Moreover, Yu et al. [59] need to fine-tune an existing model to achieve high classification accuracy and the performance of Chao et al. [12]’s method (PAIR) also depends on the choice of the evaluator LLM.
>
> **“...lower attack success rate on well aligned open-sourced model, Llama-2-7b-chat.”** Yes, our method has a lower success rate on the Llama-2-7B-chat model. But this is expected as prior methods, including PAIR, also have a low success rate with Llama-2-7B-chat. One reason for this is that Llama-2-7B-chat can be overly protective: for instance, it has been observed (see Figure 5 here https://arxiv.org/pdf/2310.08419v2) that Llama-2-7B-chat refuses to respond to benign requests if they mention certain words or phrases. Further, the fact that Llama-2-7B-chat is resilient to several jailbreaking methods can be seen as a validation that its safety training is effective.

---

> > ### Comment · Reviewer_2Heq · 2024-08-09
> >
> > Thanks for the reply. I tend to keep my score as borderline according to the following reasons.
> > 1. The dependence on a powerful and costly evaluator is one of the drawbacks of this method, others having the same problem is not a reasonable excuse for this.
> > 2. The Llama3 and Gemma2 exps are not conducted and the limitation of running time and money cost.

---

> > > ### Author Response · Authors · 2024-08-13
> > > **Thank you for your response**
> > >
> > > Thank you for taking the time to write back.
> > >
> > > Regarding the performance's dependence on the evaluator, we agree that this is a drawback. However, we do not agree that it affects the contribution of the method over existing ones, since achieving high performance with existing black-box methods (such as GPT-Fuzzer of Yu et al. [59]) requires fine-tuning existing models.
> > >
> > > Regarding the low success rate on "well-aligned open-sourced model" (e.g., Llama-2-7B model), we would like to stress that the method has high performance on many well-aligned and state-of-the-art LLMs (such as GPT-4o, GPT-3.5-Turbo, and Gemini-Pro). We do not understand why jailbreaking small open-source LLMs should be a criterion for evaluation: indeed, simple LLMs such as Llama-2-7B can be hard to jailbreak as they cannot follow complex instructions. This makes them less useful but is also one reason why they can be overly protective. This makes these models hard to jailbreak as mentioned in our earlier response.
> > >
> > > Finally, regarding the evaluations with Llama-3 and Gemma-2 as targets. We note that Llama-3 was released less than a month before the abstract submission deadline (see https://ai.meta.com/blog/meta-llama-3/) and Gemma-2 was released *after* the deadline (see https://ai.google.dev/gemma/docs/releases). This prevented us from thoroughly evaluating the method with these models. We would be happy to include these evaluations in the final version.

---

### Official Review · Reviewer_LFqH · 2024-07-11

**Soundness:** 2
**Presentation:** 3
**Contribution:** 2
**Rating:** 6
**Confidence:** 4

**Summary:**

This paper propose Tree of Attacks with Pruning (TAP) to jailbreak blackbox LLMs automatically. TAP has four steps, including branching, pruning, attack and assess, and pruning. TAP leverage an attacker (an LLM) to generate variations of the provided prompt, and the evaluator (another LLM) decide which variations to send to the target model. TAP is an advanced version of PAIR, reaching a higher success rate more effectively.

**Strengths:**

1, TAP can reach a high success rate on GPT4, GPT4o and Gemini-Pro.

2, Compared to PAIR, TAP is more efficiency on most of the black box model, suggesting the effectiveness of the branching and pruning.

3, TAP shows a better transferability between models.

**Weaknesses:**

1, As a method built on PAIR, with branching and pruning, the novelty of the method is limited.

2, In Fig.1, a typo "branching".

**Questions:**

1, In table 7, the attacker’s system prompt is long and carefully designed. The prompts try to get the LLM to make up a story for making the harmful behavior more reasonable and align with human value. This well designed prompt may be a key to success in jailbreak.  It would help if the paper includes replacing the prompt with other evaluator/attacker prompts and show some results to state the success rate comes mostly from pruning instead of prompt engineering.

2, In section 6 discussion and appendix A.2, TAP-No-Prune has a higher success rate than TAP Prune, because retains the w = 10 highest scoring prompts and deletes the rest. Does it suggest that the off-topic prompts get higher scores?

3, As the subtree in Figure 12-14, is the improvement of the attacker generated by evaluator? What's the prompts for generating these? And are the revised prompts share the same attacker's prompts as the original generation?

4, In the branching stage, does the length of the re-writing prompts affect the success rate? Perhaps adding some constraints on the generation length may also help the jailbreak success rate.

**Limitations:**

Please refer to questions.

---

> ### Author Rebuttal · Authors · 2024-08-07
>
> Thank you for your thorough review. We respond to your questions below and hope that you will strengthen your support for our submission.
>
> **“a typo "branching"”** Thanks, we will correct this.
>
> **“attacker’s system prompt is long and carefully designed…”** Yes, the attacker’s system prompt in Table 7 is long and carefully designed. It is the same prompt as used by PAIR (an existing method). We use it in our simulations to ensure that both our method and PAIR use the same system prompt. This ensures that any gains in performance compared to PAIR are due to our method and not due to the engineering of the system prompt.
>
> **“...would help if the paper includes replacing the prompt with other evaluator/attacker prompts and show some results to state the success rate comes mostly from pruning instead of prompt engineering.”** That’s a great suggestion. We implemented a variant of our method that uses a simpler system prompt: this system prompt simplifies the prompt in Table 7 by removing the detailed examples. We evaluated this variant with GPT-4-Turbo as the target and, matching our other simulations, GPT-4 as the evaluator and Vicuna-13B as the attacker.  We observe that this variant jailbreaks a significantly higher number of prompts than PAIR (82% vs 44%) with fewer queries (35.0 vs 47.1) even though PAIR uses a more sophisticated attacker prompt with detailed examples. We will include this result and the full text of the simpler system prompt in the final version.
>
> **“In section 6 discussion and appendix A.2…Does it suggest that the off-topic prompts get higher scores?”** Yes, these results suggest that off-topic prompts can sometimes get higher scores than on-topic prompts.
>
> **“is the improvement…generated by evaluator? What's the prompts for generating these?...”** Sorry for any confusion. The improvements are generated by the attacker. The attacker’s system prompt (in Table 7) is used for generating improvements. This system prompt requests the attacker to both generate an improvement and, based on the improvement, modify its previous attack.
>
>
> **“...How do generation constraints on the revision prompts affect jailbreak effectiveness?"** That is a nice suggestion, thanks. Following the suggestion, we strengthened the generation constraints from 500 tokens per revised prompt to 250 tokens per revised prompt. We evaluated this variant–which has a stricter generation constraint–with GPT-4-Turbo as the target, GPT-4 as the evaluator, and Vicuna-13B as the attacker. It achieves a similar fraction of jailbreaks (82%) as the original implementation of our method (84%) with a similar number of average queries (24.8 compared to 22.5).

---

> > ### Comment · Reviewer_LFqH · 2024-08-13
> >
> > Thanks authors for the reply. And my concerns are mainly addressed. From the reply we can see that the method is somehow not sensitive to the system prompts and the generation constraints, and I encourage the authors to include these in the final version. And I believe that this paper should be accepted.

---

### Official Review · Reviewer_psVX · 2024-07-12

**Soundness:** 3
**Presentation:** 2
**Contribution:** 3
**Rating:** 7
**Confidence:** 4

**Summary:**

This paper presents a novel jailbreaking attack based on the PAIR attack, enhanced with branching and pruning techniques. The attack generates multiple prompts through branching, then applies two pruning steps: removing off-topic prompts and eliminating prompts with low scores after testing them against the victim model. The method achieves a higher effectiveness with a lower number of queries than previous work. The study evaluates the attack's transfer performance, the effectiveness of the attack when LLaMA Guard is used as a defense, and includes an ablation study on the contribution of the different attack components.

**Strengths:**

**Attack effectiveness** The attack is strong and uses less queries than previous work. The branching and pruning steps are effective in generating effective jailbreaks.

**Thorough evaluation** The evaluation is thorough and includes transfer performance and the effectiveness of the attack against LLaMA Guard. Moreover, the ablation study provides insights into the contribution of the different attack components.

**Clear writing** The paper is well-written and easy to follow (except for the redundancy mentioned in the Weaknesses below).

**Thorough experimental details** The paper provides a lot of details about the experiments in the appendix, which is helpful for reproducibility.

**Weaknesses:**

**Redundancy in the paper content**. I find subsection 1.2 with the basic description of TAP redundant and makes the Introduction section overly long. There is a big overlap with the content of section 3. I suggest merging the two sections.

**Unclear calibration of the Judge model**. The judge model give scores that are between 1 and 10, however, in my experience while using LLMs, they are usually poorly calibrated when assigning scores. I have then two questions: 1) would it be feasible to look at how calibrated is the judge model compared to a human baseline? 2) how important is it that the model is actually calibrated when using it for the pruning step? It would be interesting to see how the pruning step would perform when selecting random prompts instead of those with the highest assigned score (except of course for the prompts that are considerered to be fully successful, with a score of 10). It seems to me that the score domain is excessively granular even for a human to assign scores, so using a coarser score scale might be more effective and meaningful. I understand that this is the same prompt as the one used in PAIR, but it seems to me that the score in TAP has a more crucial role than in PAIR.

**Questions:**

- What average is used for the number of queries in table 2? Is it the arithmetic mean? Are there many outliers? What is the median?
- See my questions about model calibration above.

**Limitations:**

The authors discuss the limitations of their work.

---

> ### Author Rebuttal · Authors · 2024-08-07
>
> Thank you for appreciating our thorough evaluation and clear writing. We take your feedback seriously and, following your suggestion, will shorten Section 1.2 to reduce the overlap with Section 3. We answer your specific questions below and hope you continue to support the paper.
>
> **“how calibrated is the judge model compared to a human baseline?”** Our judge model can indeed be miscalibrated. Unfortunately, accurately measuring the calibration seems hard as we have a small number of examples (only 50). In the spirit of comparing the judge model’s performance to a human baseline, we evaluate its false positive and false negative rates in labeling examples as jailbreaks (i.e., assigning them a score of 10): we find that its false positive and false negative rate are not too large–13% and 0% respectively (Section A.2). We will include a discussion on the potential of miscalibration and its consequences in the final version.
>
> **“how the pruning step would perform when selecting random prompts instead of those with the highest assigned score”** Thanks for the suggestion. Following it, we evaluated the variant of our method where Phase 2 of pruning randomly selects $w=10$ prompts (instead of selecting the $w$ prompts with the highest score). We selected GPT-4-Turbo as the target and, matching the simulations in the paper, selected GPT4 as the evaluator and Vicuna-13B as the attacker.  Surprisingly, this method only jailbreaks 62% of the prompts. The number of queries sent has a mean of 34.8 and a median of 24.5. This is significantly lower than the success rate of 84% of the standard variant of TAP (that retains the prompts $w$ with the highest scores) demonstrating that pruning based on the evaluator’s scores improves performance.
>
> **“It seems to me that the score domain is excessively granular even for a human to assign scores, so using a coarser score scale might be more effective and meaningful.”** Thank you for the great suggestion. Following your suggestion, we evaluated a variant of TAP where the evaluator uses a coarser score scale, namely, binary scores. We fix GPT-4-Turbo as the target, GPT4 as the evaluator, and Vicuna-13B as the attacker. We find that this improves the success rate from 84% to 86% while sending a similar number of queries (23.4 with binary score scale vs 22.5 with finer score scale). We will include these results in the final version of the paper.
>
> **“What average is used for the number of queries in table 2? Is it the arithmetic mean? Are there many outliers? What is the median?”** Yes, the average in Table 2 is the arithmetic mean. The medians of the queries are close to the arithmetic means, as shown below:
> |                  | Vicuna | Llama-7B | GPT3.5 | GPT4 | GPT4-Turbo | GPT-4o | PaLM-2 | Gemini-Pro | Claude3 Opus |
> |------------------|--------|----------|--------|------|------------|--------|--------|------------|--------------|
> | Jailbreak %      | 98%    | 4%       | 76%    | 90%  | 84%        | 94%    | 98%    | 96%        | 60%          |
> | Mean # Queries   | 11.8   | 66.4     | 23.1   | 28.8 | 22.5       | 16.2   | 16.2   | 12.4       | 116.2        |
> | Median # Queries | 8.0    | 79      | 18.0   | 23.5 | 23.0       | 14.5    | 9.0    | 8.0        | 120          |

---

> > ### Comment · Reviewer_psVX · 2024-08-09
> >
> > Thank you for your reply. I am glad that some of my suggestions turned out to be useful, and I believe that this paper should be accepted.

---

### Decision · Program_Chairs · 2024-09-25

**Decision:**

Accept (poster)

**Comment:**

This paper presents an efficient and effective jailbreaking attack against LLMs. The importance of this work is in demonstrating the limitations of existing LLM guardrails. Given the large amount of attention and resources that have gone into LLM alignment, work like this is likely to have a high impact in the short term, impacting the direction of future LLM research. Reviewers have a few suggestions for improvements, which the authors have already responded to.